# Water chlorination increases the relative abundance of an antibiotic resistance marker in developing sourdough starters

Pearson Lau,[1] Swapan Jain,[2] Gabriel G. Perron[1,3,4]

**ABSTRACT**   Multiple factors explain the proper development of sourdough starters. Although the role of raw ingredients and geography, among other things, have been widely studied recently, the possible effect of air quality and water chlorination on the overall bacterial communities associated with sourdough remains to be explored. Here, using *16S rRNA* amplicon sequencing, we show that clean, filtered-air severely limited the presence of lactic acid bacteria in sourdough starters, suggesting that surrounding air is an important source of microorganisms necessary for the development of sourdough starters. We also show that water chlorination at levels commonly found in drinking water systems has a limited impact on the overall bacterial communities developing in sourdough starters. However, using targeted sequencing, which offers a higher resolution, we found that the abundance of integron 1, a genetic mechanism responsible for the horizontal exchange of antibiotic-resistance genes in spoilage and pathogenic bacteria, increased significantly with the level of water chlorination. Although our results suggest that water chlorination might not impact sourdough starters at a deep phylogenetic level, they indicate that it can favor the spread of genetic elements associated with spoilage bacteria.

**IMPORTANCE**  Proper development of sourdough starters is critical for making tasty and healthy bread. Although many factors contributing to sourdough development have been studied, the effect of water chlorination on the bacterial communities in sourdough has been largely ignored. Researchers used sequencing techniques to investigate this effect and found that water chlorination at levels commonly found in drinking water systems has a limited impact on the overall bacterial communities developing in sourdough starters. However, they discovered that water chlorination could increase the abundance of integron 1, a genetic mechanism responsible for the horizontal exchange of antibiotic resistance genes in spoilage and pathogenic bacteria. This suggests that water chlorination could favor the growth of key spoilage bacteria and compromise the quality and safety of the bread. These findings emphasize the importance of considering water quality when developing sourdough starters for the best possible bread.

**KEYWORDS**   sourdough, food microbioloy, *16S rRNA* Amplicon, integron 1, starters, microbiomes, fermentation

T he use of sourdough starters is considered one of the great advancements in cooking (1). As the primary bread-leavening agent until the European Industrial Revolution (2), sourdough starters increased the nutritional value of bread and made the latter a staple food in many parts of the world (3). More recently, sourdough regained popularity and has increasingly been celebrated for its desirable gastronomic properties (4, 5)

Traditional sourdough starters are made from a mixture of flour and water fermented naturally by diverse populations of yeast and bacteria (6). As the starter ferments,

Address correspondence to Gabriel G. Perron, gperron@bard.edu.

The authors declare no conflict of interest.

microbial populations that are initially diverse become quickly dominated by lactic acid bacteria and, to a lesser extent, by acetic acid bacteria (3, 7). The dominant presence of lactic acid bacteria results in chemical, metabolic, and enzymatic activities that not only increase the nutritional value of sourdough bread but also inhibit the growth of other bacterial genera, contributing to the self-preserving properties of sourdough (8, 9).

Multiple factors explain the proper development of microbial communities in sourdough starters (10). For example, sourdough starters are heavily influenced by temperature variations (7) and possibly by the presence of microorganisms in the air surrounding the sourdough starters (11, 12). Crucially, the ingredients used in starter generation also play a defining role in shaping sourdough structure (13, 14). However, the role of water quality, the other main ingredient in sourdough starter preparation, has only recently been recognized as a potential factor shaping microbial communities in sourdough starters (15). More specifically, the presence of disinfectant residuals commonly used in drinking water is widely believed among professional bakers to negatively impact the proper development of microbial communities found in sourdough starter, potentially changing the flavor profiles of the sourdough (11, 16).

Chlorination is the most common disinfectant used in public water distribution systems (17). Most often, water chlorination is achieved by adding sodium hypochlorite, which leads to the presence of hypochlorite ion (OCl-) in the medium, inhibiting bacterial growth by disrupting metabolism and enzymatic inactivation (18). The efficiency with which chlorination can stop the spread of most water-borne pathogens is considered one of the most outstanding achievements in public health of the past century (19). However, the inhibiting activity of chlorine present in water is not limited to pathogenic bacteria (20, 21). Indeed, the potential impact of chlorine is predicted to extend to most microbial communities exposed to chlorinated water (22).

The presence of free chlorine in the water could alter the chemical properties of sourdoughs. For example, hypochlorite ion (OCl-) is an oxidizing agent that can break glycosidic bonds within bread starches, reducing the gluten network and subsequently reducing the ability of flour components to gel together during the baking process (23). Water chlorination can also reduce flour's lipid content due to the formation of chlorine derivatives (24). In addition to affecting the gustatory properties of the sourdough, such changes could affect developing microbial communities.

Chlorination was also demonstrated to promote the spread of antibiotic resistance (25, 26). Although chlorination contributes to reducing the number of antibiotic-resistant bacteria in treated water at first (27, 28), the continued presence of low concentrations of free chlorine in water can select for antibiotic-resistant bacteria downstream from treatment plants (29) and promote the exchange of antibiotic resistance genes among bacteria via horizontal gene transfer (25, 26). Although the presence of antibiotic-resistance genes in sourdough and sourdough starters is unlikely to be a major health concern, even though DNA is not destroyed during cooking (30), antibiotic resistance is often associated with spoilage bacteria (31). Therefore, selecting for antibiotic-resistant bacteria in sourdough could affect the bread's quality and preservation.

Here, using *16S rRNA* amplicon sequencing, we investigate the effect of chlorinated water on the development of bacterial communities in sourdough starters. In addition, we monitor the possible effect of water chlorination on the spread of integron 1, an important genetic element associated with the spread of antibiotic resistance in pathogenic bacteria (32). Although we show that water chlorination has a limited impact on the overall bacterial community structure developing in sourdough starters, we found that chlorinated water increased the abundance of integron 1, an indicator associated with clinically important antibiotic resistance genes, pathogenic bacteria, and spoilage bacteria.

## MATERIALS AND METHODS

### Establishment of sourdough starters

Sourdough starters are composed of two ingredients, flour and water, mixed together and regularly replenished to favor microbial growth. To control for possible variation in flour composition, we used a single bag of organic, stone-ground whole wheat flour (King Arthur Flour, Norwich, VT) for the entire experimental period of sourdough fermentation. According to the manufacturer's website, the flour is made from dark northern hard red wheat, a varietal of the common wheat (*Triticum aestivum*) that contains a higher protein content (~13.8%). We established the sourdough starters by combining 10 g of flour with 10 mL of control or treated water (see below) in sterilized polypropylene Nalgene bottles. Next, we mixed manually with an ethanol-sterilized glass rod, resulting in a dough yield (DY) of 200, or pastelike 'consistency' (33).

We then fed the sourdough starter every 24 hours (±2 hours), commonly called "backslopping," by discarding 50% of the initial DY, and replenishing with a fresh mixture of flour and water to achive the initial DI. We ensured the complete homogenization of each sourdough starter by pouring the dough into a sterile bag and homogenizing it using the BagMixer 5000 (Interscience, Saint Nom la Brétèche, France) using default parameters for 60 seconds. We then remove 50% of the starter by weight and replace it with a fresh mixture of flour and water, as described above. Next, the dough was scraped down and remixed on the same settings. The freshly fed starter paste was then squeezed into a new sterilized polypropylene bottle. The bottles were placed in a dark cupboard for 24 hours at ambient room temperature maintained at 22–25°C. We repeated the feeding procedure six times for a total fermentation time of 7 days.

To identify the optimal growing conditions for investigating the possible effects of water chlorination, we established two independent trials. First, we chose to limit the exposure of the sourdough starters to bacteria in the air by having an "air-tight" container or screwing on the lid tightly. The only times the container lids were removed were to refresh the starter. Second, we conducted a second set of experiments with identical conditions, but this time allowing exposure to air located in a working kitchen. For each experiment, we established three replicate starters for each control and chlorine treatment for a total of 18 experimental populations. All measurements and mixing were done under sterile, aseptic conditions throughout the study.

### Water chlorination treatments

To test for the possible effects of water chlorination on sourdough starters, we established and maintained sourdough starters with three water chlorination treatments resulting in three concentrations of free chlorine in the water: 0 ppm (or control); 0.5 ppm (0.5 mg/L), and 4.0 ppm (4.0 mg/L). The treatments were chosen to reflect the minimum and maximum residual amount of chlorine in finished drinking water in the United States of America (CDC, 2020). We prepared chlorinated waters daily before sourdough feeding by diluting sodium hypochlorite, or NaOCl, into 100 mL of sterilized water.

We tested for the water chlorination level in each water preparation using the N, N-dimethylacetamide method as implemented using the LaMotte Chlorine test kit (LaMotte, Baltimore, USA). Briefly, 5 mL of chlorinated water was mixed with N, N diethyl-p-phenylenediamine and was compared with the provided color chart, indicating the available chlorine concentration of the solution. Because we were also concerned with the presence of other organic material in the water interfering with the disinfection efficacy of chlorine concentrations in our treatments, we tested the residual chlorine concentrations using the digital colorimeters method as described in the CDC protocol for measuring residual free chlorine (34). We confirmed that residual chlorine concentration was stable throughout our experiment.

## Sample processing

We used a *16S rRNA* amplicon sequencing approach to characterize the bacterial communities found in each sourdough starter, also known as food microbiomes. We first extracted the bacterial DNA from 1 g of dough, which we diluted in 9 mL sterile, peptone physiological solution (0.1% peptone, 0.85% NaCl). We then extracted microbial DNA from the diluted starter using the procedure described in the MoBio PowerFood DNA Extraction kit (MoBio, Carlsbad, CA). Finally, we amplified the V4 region of the *16S rRNA* gene using the Golay-barcoded primers 515F and 806R (35). Following gel purification, libraries were pooled at equimolar ratios and sequenced on the MiSeq paired-end Illumina platform adapted for 250 bp paired-end reads (Wright Labs, Huntingdon, PA) according to the Earth Microbiome Project's protocol (36). All unprocessed sequence reads are available at the Sequence Read Archive of the National Center for Biotechnology Information (NCBI accession number: PRJNA784321).

## Processing of *16S rRNA* amplicon sequence data

We characterized the microbiomes of each sourdough starter sample by identifying and tabulating the number of different sequence variants, also known as amplicon sequence variants (i.e., ASVs). Sequence variants can then be assigned to a taxonomic rank, usually at the genus level, providing additional information about the biology of each microbiome community. More specifically, we processed the *16S rRNA* reads using the *DADA2* pipeline version 1.20 (37) available at (https://github.com/benjjneb/dada2) using standard parameters unless specified and implemented in *R* version 4.1.1 (http://www.r-project.org) (see (38) for full details). In total, we obtained a total of 1,160,000 pairs of forward and reverse reads, (excluding eight samples that failed to sequence) with an average read length of 250 base pairs, totaling ~583 G bases, and an average sequencing depth per sample of 41,642.9 paired-reads. Each sequence read was then quality-checked, trimmed (i.e., forward reads at 240 bp and reverse reads at 225 bp), assessed for chimeric contaminants, and de-noised for possible sequencing error. Following quality filtering, we conserved 980,261 (84.1% of the initial) paired-end reads.

Taxonomy was assigned using both the DADA2 native taxa identifier function as well as *IDTAXA* (39) available via the *DECIPHER* Bioconductor package (DOI: 10.18129/B9.2bioc.DECIPHER) trained on the SILVA ribosomal RNA gene database version 138.1 (40) as well as the RDP trainset 18 (41, 42). A complete list of all ASVs and their abundance in each sample can be found in Table S1, and a complete taxonomic assignment can be found in Table S2. Finally, we build a maximum likelihood phylogenetic tree based on multiple alignments of all the ASVs using the *phangorn* package version 2.1.3 (43); the latter is used to estimate the total phylogenetic, or evolutionary, distance present in each sample.

## Microbial community analysis

Microbiomes' diversity was analyzed using *phyloseq* version 1.30.0 (available at https://joey711.github.io/phyloseq/) implemented in *R* and visualized in *ggplot2* (44). A mapping file linking sample names and the different treatments is provided in Table S3. To estimate diversity indices, we rarefed all samples to the lowest sampling depth and estimated the total number of ASVs, Chao1, which is the predicted number of ASVs in the whole sample, as well as diversity as Simpson's Index, that is, *D*. Although richness considers the total number of ASVs, diversity includes evenness measures among the different ASVs present in a sample. We tested whether richness and diversity indices differed among treatments using linear modeling and comparing the different statistical models with Akaike's Information Criterion (AIC) as implemented in *R's* stats package.

To test whether there were statistical differences in population structure between treatments (e.g., control microbiomes vs. microbiomes exposed to chlorinated water as well microbiomes exposed to air vs. microbiomes not exposed to air), we performed

Principal Coordinate Analyses (PCoA) on phylogenetic distances calculated as weighted UniFrac distance scores (45). We used a permutational multivariate analysis of variance (PERMANOVA) implemented via the *adonis* function (46) of *vegan* version 2.5.6 to test for significance. The latter is a non-parametric method that estimates *F*-values from distance matrices among groups and relies on permutations to determine the statistical significance of observed differences among group means. Finally, we confirmed that each test respected the homogeneity of variances assumption using the betadisper method of the *vegan* package (46).

## Quantify the presence of an antibiotic resistance and spoilage bacteria marker

Finally, we investigated whether chlorinated water increased the relative abundance of *intI1*, a gene encoding integron class 1 integrase (32). The latter is almost entirely associated with spoilage or potentially harmful bacteria and facilitates the spread of antibiotic-resistant genes among bacteria (47, 48). As described elsewhere (47), we quantified *16S rRNA* and *intI1* gene copy numbers from triplicate reactions using the Bio-Rad CFX96 Real-Time PCR Detection System (Bio-Rad Laboratories, Hercules, CA, USA). We included internal standard curves with each qPCR run to estimate the copy number for both *intI1* and *16S rRNA*. To normalize our *intI1* findings and allow comparisons between samples, we divided the *intI1* copy number by the *16S rRNA* copy number to provide us with a measure of *intI1* relative abundance in each sample. The *16S rRNA* copy number was adjusted by dividing it by 4.2, the average number of *16S rRNA* copies in each bacteria cell (49). Finally, we used linear modeling to test for the effect of water chlorination on *intI1* relative abundance and compared the different statistical models with Akaike's Information Criterion (AIC) as implemented in *R's* stats package. Although we used the square root-transformed data for statistical analysis, we plotted the raw data.

## RESULTS

### Overall microbial population structure

Using 16S *rRNA* amplicon sequencing to characterize the sourdough starter microbiomes, we identified a total of 150 unique ASVs with a median of nine ASVs found in each starter (Table S1). The number of ASVs found in each starter varied greatly from four ASVs found in a starter fermented for 1 day to up to 65 ASVs in a starter fermented for 7 days. In total, ASVs were matched to 82 different bacteria genera (Table S2), with *Latilactobacillus* being the most common among our samples, 49.4(39.5)%, followed by *Pantoea* sp., 27.2(39.0)%, *Weissella* sp., 9.1(23.2)%, and *Pseudomonas* sp., 8.7(26.7)%. As taxonomy changed slightly depending on the database and algorithm used (Fig. S1), we will present our results using the RDP database with the *TaxaID* taxonomy identifier thereafter; the latter offered the smallest number of unidentified ASVs.

### Identifying the source of microbial fermentation in new sourdough starters

We first wanted to identify the ideal condition to establish sourdough starters in our facilities. To do so, we established two sourdough starter trials: one under sterile laboratory conditions with filtered air and the second in a kitchen environment with exposure to unfiltered air. When comparing the microbial communities found in the sourdoughs after 7 days of fermentation and using the same feeding procedure, we found a significant divergence in the population structure estimated from the weighted UniFrac distance matrix, based on phylogenetic distance, between the two groups ($F_{(1,16)}$ = 27.7; $R^2$ = 0.65; adj-$P$ = 0.002; Fig. 1a). Similarly, we also found a significant difference in population structure when comparing the overall dissimilarity in ASV composition between the two groups ($F_{(1,23)}$ = 15.57; adj-$P$ = 0.002; Fig. 1b). Although the above study could be influenced by a certain degree of heterogeneity in variance between the two groups, we found similar results even when the number of samples was kept constant

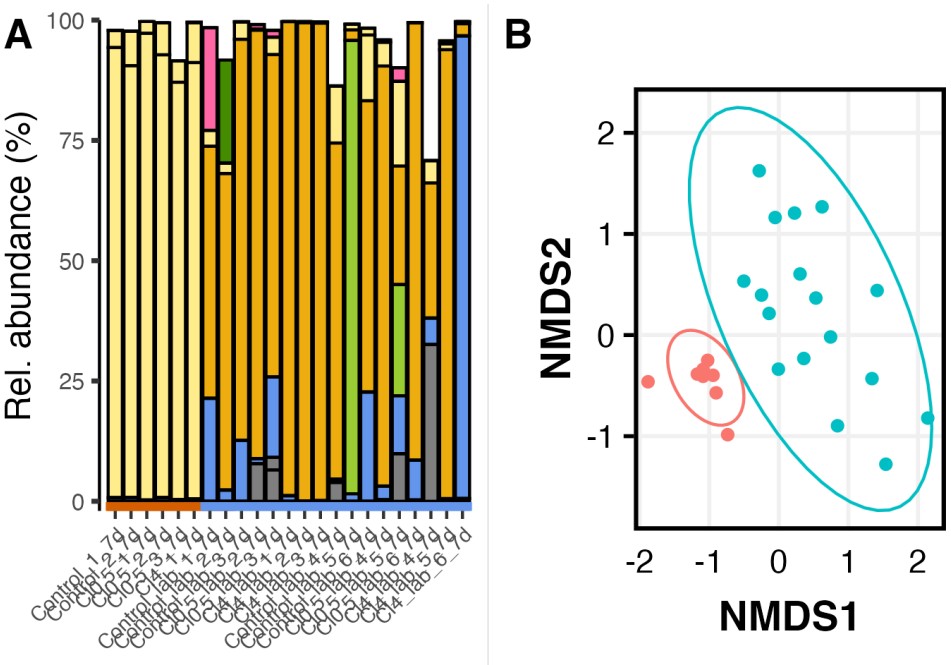

**FIG 1** Microbial diversity found in sourdough starters in unfiltered air (red) and filtered air (blue). (A) Relative abundance of the six most common bacteria genera: *Enterococcus* (green), *Klebsiella* (pink), *Pantoea* (corn), *Latilactobacillus* (dark gold), *Pseudomonas* (olive), *Erwinia* (cornflower blue), and unknown (Gray). (B) Differences in population structure estimated from non-metric multidimensional scaling of distance matrix calculated from the Bray-Curtis dissimilarity index. Bacteria types found in Sourdough fermented in the presence of unfiltered and filtered air were more clustered than expected by chance alone. Sourdough starters fermented in fileted air are shown in blue, whereas starters fermented in unfiltered air are presented in red.

between the two groups by subsampling, suggesting that there is a real divergence in population structure.

Although we did not find significant differences in the number of observed ASVs ($F_{(1,15)}$ = 5.08; adj-$P$ = 0.16; Fig. S2A) or evenness measured as the Simpson's Index ($F_{(1,25)}$ = 4.16; adj-$P$ = 0.18; Fig. S2B), the main difference we observed between the two trials was that the most common ASV found in the presence of unfiltered air was identified as *Latilactobacillus* sp. (Fig. 1a). In contrast, the most common ASV found under clean laboratory conditions was identified as *Panteoa* sp., a genus not commonly associated with sourdough starters. Although the two groups showed some level of heteroscedasticity ($P$ = 0.04), we found that the significant difference in community structure between communities exposed to unfiltered air and those exposed to laboratory-filtered air helped after subsampling the latter. Finally, we found that the starters grown in laboratory conditions were significantly less likely to result in communities where lactic acid bacteria made up at least 80% of the total identified ASVs (Fisher's exact test: $P$ = 0.05). For the above reasons, we decided to investigate the possible effect of water chlorination on sourdoughs exposed to air only.

## The effect of water chlorination on sourdough starters

To test for the possible effect of water chlorination on the microbial communities developing in sourdough starters, we exposed starters to three different chlorination treatments, including a control group not exposed to hypochlorite ion. We found no evidence that water chlorination affected the overall structure of microbial communities in sourdough starters. Overall, the same few ASV dominated the populations by day 7 in all treatments (Fig. S3A). Using principal component analysis to detect possible changes in community structure measured as phylogenetic distance via Unifrac scores, we found

that microbial populations changed significantly over time (ADONIS: $F_{(1,15)}$ = 8.01; $R^2$ = 0.35; adj-$P$ = 0.004) similarly in all water chlorination treatments (ADONIS: $F_{(2,14)}$ = 0.12; $R^2$ = 0.02; adj-$P$ >0.99; Fig. S3B). In other words, even a chlorine concentration at a level observed in some of the most chlorinated public water systems, that is, 4 ppm, did not modify the relative abundance of most ASVs and taxa observed in the different samples.

Similarly, although the number of predicted ASVs observed in the starters decreased over time ($F_{(2,5)}$ = 0.10; $P$ = 0.90; Fig. 2a), we found that the number of ASVs did not differ between the water chlorination treatments ($F_{(2,5)}$ = 0.10; $P$ = 0.90; Fig. 2a). We found that chlorine levels did not affect diversity as measured by Simpson's index ($D$), a measure that is sensitive to how evenly the ASVs are distributed in the samples ($F_{(2,5)}$ = 0.02; $P$ = 0.98; Fig. 2b). Similar to observed ASVs, however, diversity decreased over time ($F_{(1,5)}$ = 11.92; $P$ = 0.02; Fig. 2b). Interestingly, when we look at the dispersion in the number of ASVs around the mean for each treatment, as measured by variance, we note that variance seems to increase in the presence of chlorine. Unfortunately, we do not have enough data points to test for this pattern. Still, this result suggests that water chlorination could result in finer changes while not changing the core microbial communities in the starters.

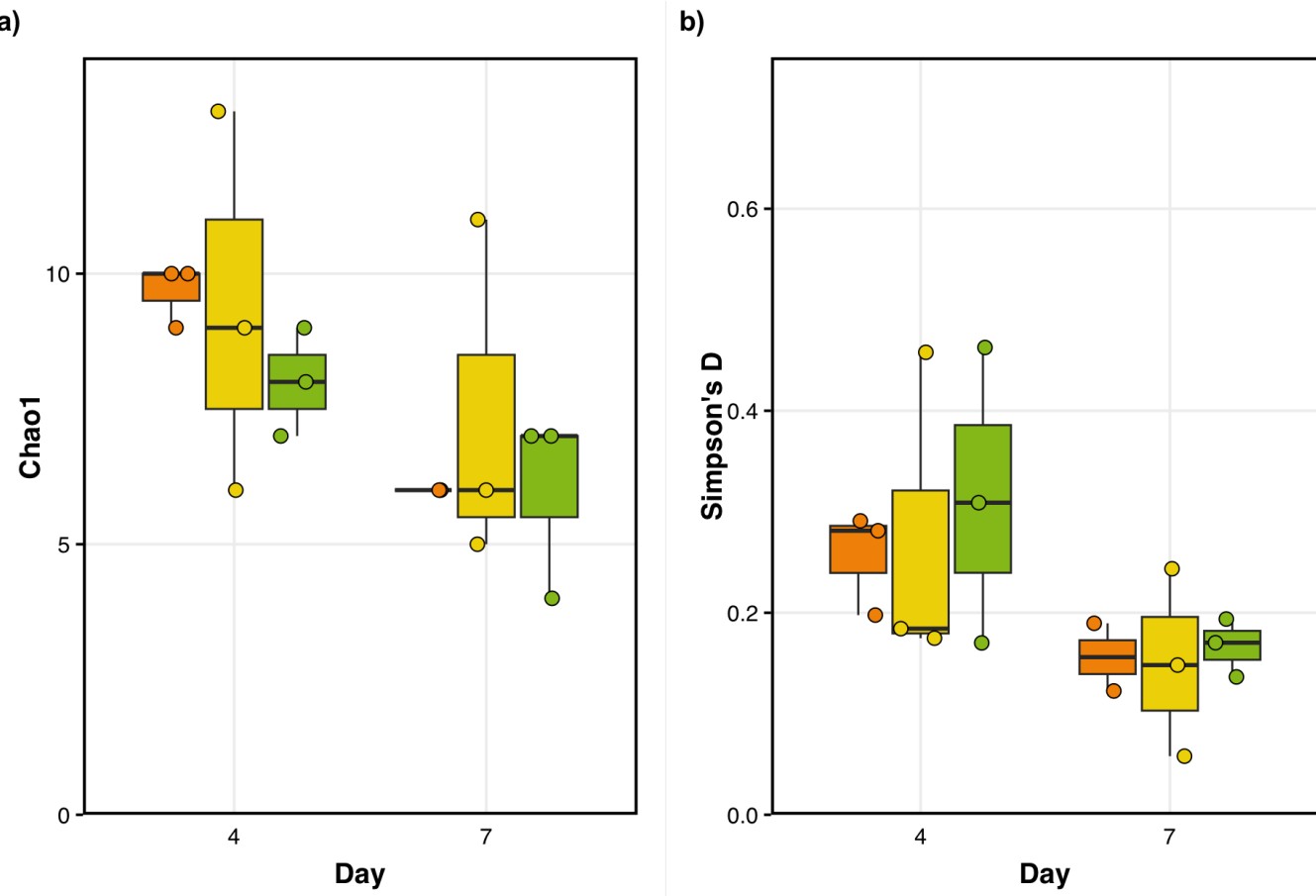

**FIG 2** Effect of water chlorination on bacterial diversity found in sourdough starters. (a) Although the number of predicted ASVs, estimated as Chao1, decreased from day 4–7 ($F_{(1,5)}$ = 15.32; $P$ = 0.011), the number of ASVs did not change between free chlorine concentrations ($F_{(2,5)}$ = 0.08; $P$ = 0.92) . (b) Similarly, diversity, measured as Simpson's $D$, decreased over time ($F_{(1,5)}$ = 11.92; $P$ = 0.018) but was similar among free chlorine concentrations ($F_{(2,5)}$ = 0.01; $P$ = 0.99). Concentrations of chlorine in the water tested were 0 mg/L, shown in orange, 0.5 mg/L in yellow, and 4 mg/L in green. Box plots show the median as well as the interquartile range. Individual points represent the score for each sample.

## Investigating the presence of integron 1 during fermentation

Although we did not observe major changes in community structure or diversity, our previous results suggest that changes in community dynamics could have happened at a finer scale (38, 50, 51). For example, it is possible that chlorine could exert selective pressures on a resistant strain within a genus or even at the gene level via horizontal gene transfer. For this reason, we investigated the relative abundance of *intl1*, the gene encoding for integron class 1. The latter is a genetic mechanism enabling the quick transfer of genes and is almost always associated with bacteria with a spoiling or pathogenic potential and is usually associated with antibiotic resistance (32, 48, 52).

Using quantitative PCR, we found that chlorinated water affected the relative abundance of *intl* in sourdoughs over time (treatment:time: $F_{(2,18)} = 4.17$; $P = 0.03$; Fig. 3b). More specifically, we found that the highest chlorine concentration, that is, 4 ppm, significantly increased the relative abundance of *intl1* by day 7 ($t = 2.59$; $P = 0.02$; Fig. 3a). Interestingly, we found no difference in *intl1* relative abundance among the different chlorine concentrations at day 4, suggesting that selective pressures for *intl1* could not be detected at this point. In other words, a larger proportion of the bacteria detected via qPCR harbored the *intl1* gene by the end of the experiment only at the highest chlorine concentration. In contrast, the relative abundance of the gene stayed more or less constant across all other treatments over time.

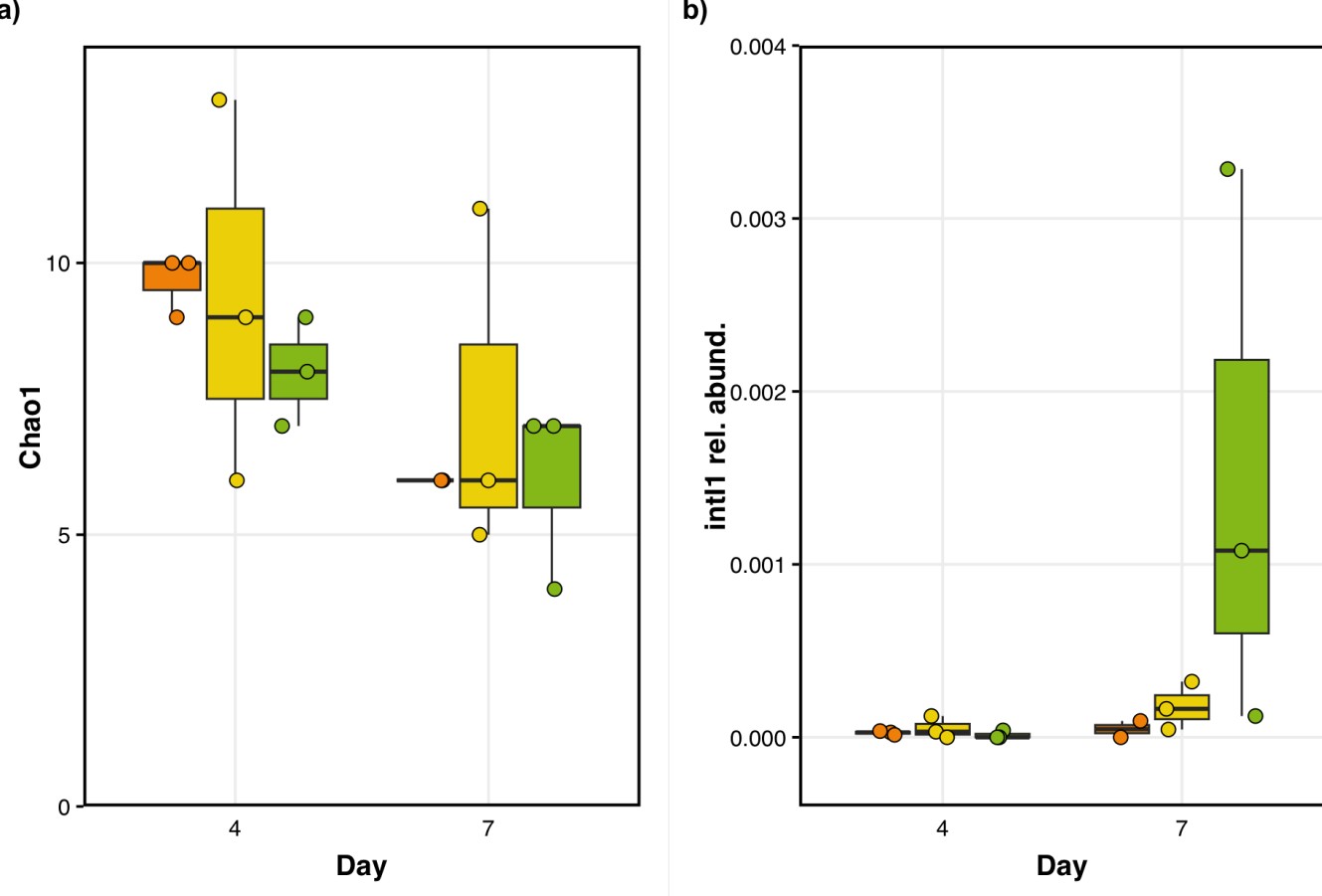

**FIG 3** Effect of water chlorination on the relative abundance of intl1 in sourdough starters. a) Chlorination affected *intl1*'s relative abundance, calculated as the proportion of *intl1* copy numbers per *16S rRNA* copies (treatment:time: $F_{(2,18)} = 4.17$; $P = 0.03$). Concentrations of chlorine in the water tested were 0 mg/L, shown in orange, 0.5 mg/L in yellow, and 4 mg/L in green.

## DISCUSSION

Multiple factors, such as flour quality or fermentation time, explain the proper development of sourdough starters (10, 15, 53). Understanding the factors that influence and shape the development of sourdough starters is not only important for the reliable production of quality sourdough but also can help us shed new light on the cultural importance of bread making. Here, we show how air quality and water chlorination can influence the development of microbial communities found in sourdough starters. We also found that sourdough starters affected by a chemical stressor like free chlorine present in water can be less resilient to the presence of possible food-spoiling bacteria or pathogenic bacteria.

More specifically, we found that sourdough starters exposed to unfiltered air developed healthy microbial communities with dominant bacteria taxa most often associated with traditional sourdough fermentation. On the other hand, when sourdough starters were exposed to filtered air in a controlled laboratory setting, the starters were dominated by *Panteoa* sp., an Enterobacteriaceae which was initially isolated as a plant pathogen and that can also be isolated from human and animal gut as well as spoiled soil and water (54). Some species of *Panteoa* are known contaminants of sourdough and can negatively affect fermentation (7, 55). The fact that *Pantoea* dominated all of our starter replicates grown independently suggests that the possible contaminant was likely present in the flour we used to establish the starters (55) and that exposure to unfiltered and well-oxygenated air is crucial for the proper development of healthy sourdough starters.

Interestingly, the amount of chlorine present did not affect the overall microbial community structure in healthy sourdough starters. Regardless of the chlorine concentration used, the same dominant bacteria taxon, that is, *Latilactobacillus* sp., was detected in all starters by the end of the experiment. The latter is a lactic acid bacterium commonly identified in sourdough starters (56, 57) and other fermented products (58, 59). In fact, *Latilactobacillus* sp. accounted for more than 80% of the total read count in all our sourdough starters, confirming that chlorine did not affect our ability to produce healthy sourdough starters as previously predicted.

Our results, however, show that chlorine could affect microbial communities at the gene level. We found that the relative abundance of the gene *intI1* increases significantly with chlorine concentration in water. More specifically, the relative abundance of *intI1* was significantly higher in starters exposed to the highest free chlorine level by the end of the feeding period. Interestingly, we did not observe this difference after 4 days of feeding the sourdough or between the other concentrations of free chlorine. This result suggests that the effect of free chlorine accumulates over time and is only detectable at higher concentrations found in public water systems. Although the copy number of *intI1* was relatively low compared with the total number of *16S rRNA* gene copies identified in our study, the presence of a marker associated with antibiotic-resistance genes and spoilage bacteria should be taken seriously. Whether our observation of this gene marker translates into the actual presence of antibiotic-resistant bacteria in sourdough remains to be tested. However, it is known that some bacterial strains associated with sourdough fermentation show intrinsic resistance to antibiotics (60) and that farming systems where raw ingredients were grown can also contribute to the presence of antimicrobial resistance composition (61). Finally, even if the genomic content of starter cultures does contain a high abundance of antibiotic-resistance genes, how it affects subsequent functionality in fermented food has yet to be determined (62).

In conclusion, our study provides an important proof-of-principle of the possible effect of water chlorination on sourdough starters and contributes to the growing body of literature investigating how environmental variables shape fermented foods. Our findings also suggest that whole-genome sequencing conducted at the population level, sometimes referred to as metagenomic, might be required to fully understand the finer changes in microbial communities impacted by water chlorination and other environ-

mental factors, possibly impacting the desired gastronomic properties of sourdough bread.

## ACKNOWLEDGMENTS

The authors would like to thank Maureen O'Callaghan-Scholl and Chef Rei Peraza for their technical assistance in the laboratory and the kitchen.

This work did not receive any specific grant from funding agencies in the public, commercial, or not-for-profit sectors.

## AUTHOR AFFILIATIONS

[1]Department of Biology, Bard College, Annandale-on-Hudson, New York, USA

[2]Department of Chemistry and Biochemistry, Bard College, Annandale-on-Hudson, New York, USA

[3]Center for Environmental Sciences & Humanities, Bard College, Annandale-on-Hudson, New York, USA

[4]Center for Genomics and Systems Biology, New York University, New York, New York, USA

## AUTHOR ORCIDs

Pearson Lau  http://orcid.org/0009-0003-7016-3008
Swapan Jain  http://orcid.org/0000-0002-7475-6754
Gabriel G. Perron  http://orcid.org/0000-0003-3526-5239

## AUTHOR CONTRIBUTIONS

Pearson Lau, Conceptualization, Formal analysis, Investigation, Methodology, Writing – original draft, Writing – review and editing.

## ADDITIONAL FILES

The following material is available online.

### Supplemental Material

**Figure S1 (Spectrum01121-23-s0001.tif).** Comparing taxonomic assignment based on different databases.
**Figure S2 (Spectrum01121-23-s0002.tif).** Effect of air exposure.
**Figure S3 (Spectrum01121-23-s0003.tif).** Effect of chlorination on bacterial community structure.
**Supplemental Figure Legend (Spectrum01121-23-s0004.docx).** Legends for Fig. S1, S2, and S3.
**Table S1 (Spectrum01121-23-s0005.csv).** List of all ASV distribution in all samples.
**Table S2 (Spectrum01121-23-s0006.csv).** List of taxonomic assignment for all ASVs.
**Table S3 (Spectrum01121-23-s0007.csv).** Metadata associated for each sample.

### Open Peer Review

**PEER REVIEW HISTORY (review-history.pdf).** An accounting of the reviewer comments and feedback.

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
