## [Reviewer comments · Microbiology Spectrum]

Microbiology Spectrum

Water chlorination increases the relative abundance of an antibiotic resistance marker in developing sourdough starters

Pearson Lau, Swapan Jain, and Gabriel Perron

Corresponding Author(s): Gabriel Perron, Bard College

Review Timeline:

Submission Date:	March 14, 2023
Editorial Decision:	July 16, 2023
Revision Received:	July 19, 2024
Accepted:	August 13, 2024

Editor: Kate Howell

Reviewer(s): The reviewers have opted to remain anonymous.

Transaction Report:

DOI: <https://doi.org/10.1128/spectrum.01121-23>

July 16, 2023

Dr. Gabriel G Perron
Bard College
Biology
30 Campus Road
Annandale-On-Hudson, NY 12504

Re: Spectrum01121-23 (Water chlorination increases the relative abundance of an antibiotic resistance marker in developing sourdough starters)

Dear Dr. Gabriel G Perron:

Many thanks for your patience as we found expert reviewers to assess your manuscript. You can see from the comments below that there are questions regarding the methodology and result interpretation that are important to address.

Link Not Available

Sincerely,

Kate Howell

Journals Department
Reviewer comments:

Reviewer #2 (Comments for the Author):

The manuscript (Spectrum01121-23) investigated the effects of both surrounding air and water chlorination on bacterial communities during the development of sourdough starters. The abundance of integron 1 gene was quantified by a qPCR method in order to discuss the horizontal exchange of antibiotic-resistance genes. Some findings were introduced in the results section. For example, the tested chlorination levels within the United States of America regulation have limited impacts on sourdough bacterial community structures. This finding seems to be informative for bakers making sourdough there.

Despite the potentially interesting findings, the conclusion is not fully supported by the data. In abstract, authors mentioned that this study indicated that water chlorination can favor the growth of key spoilage bacteria, but these key spoilage bacteria did not be explained taxonomically by 16S amplicon sequencing analysis. Thus, the authors need to show additional data to substantiate their claims. In addition, I think it is better to mention the effects of surrounding air on the bacterial community in abstract section.

My comments and suggestion on several important points are below.

1. Materials and methods section

Line193. Does the Nano seq paired-end Illumina platform actually exist? I cannot find this platform in Illumina website.

<https://jp.illumina.com/systems/sequencing-platforms.html>

Line248. Why the int1 gene abundance was divided by the adjusted 16S rRNA copy number? I did not understand the meaning of the value. Does the value try to estimate int 1 gene abundance per one bacterial cell? Please explain it more detail. Thus, I cannot evaluate the results and discussions of the qPCR analysis.

2. Results section

Line 282 What is the values of ADONIS: $F(1.16) = 27.7$, $R^2=0.65$, $P=0.65$, respectively? Please explain the meanings of these values in Materials and Methods section more detail.

Line 290. What is the meaning of "invasion"? Did the authors think lactic acid bacteria are outcompeted by the invaded bacteria? If so, please explain it more detail and discuss the mechanistic reason in discussion section.

Figure 1a. The relative abundance is usually shown by 100% bar chart. But, many bar charts seemed not to be 100%. Why?

Line 311. I cannot find figure 2A and 2B.

Line 319. Is the unit of "µg/L" true?

Line336. Please add the reference on your previous works.

Line355. The results of int1 relative abundance at the day 4 are interesting. Please discuss the reason why there were no difference among different chlorine conc.

Line391. Please add the reference on the previous prediction.

3. Discussion section

Line410. I cannot understand why the whole genome sequencing is useful to evaluate the effect of water chlorination. Can this technique be applied to an environmental survey of sourdough ecosystem? I guess this technique is usually used for genome sequence of a microbial strain which is purely cultured. Please explain it more detail.

Reviewer #3 (Comments for the Author):

Dear authors,

thank you for this very interesting read. You have conducted a nice study on the effect of water chlorination on sourdough starters.

Whilst I liked the manuscript overall, there are multiple points that need to be addressed, most of them related to the presentation of the results. I hope that you will find these comments helpful to improve your manuscript. Please find those comments below. One thing though: It is indicated in the text that you also have supplemental tables, which I, however, could not find anywhere. Have they been forgotten in the submission process?

Material and Methods:

- Line 177: What colour chart? Was this a commercial kit?

- Line 203pp: Please state what parameters were used for the dada2 and following analyses. E.g., were standard parameters used or did you modify them in any way?

- Line 218pp: Did you use rarefaction or any other means of normalization to counteract the potential effect of uneven read-counts on diversity estimates?

Results

- Line 264-266: This information about trimming and removal of chimeric reads should go to the material and methods section (see previous comments about analysis parameters).

- Line 279: What was the idea behind limiting air exposure? Did you want to avoid bacteria from the air getting into your starter? Or was your main target the exclusion of oxygen from the incubation? This should be stated somewhere. In line 289, you refer to "sterile conditions" so was your target the exclusion of microbes from the air?

- Figure 1a: Please include a figure legend which maps the bar colours to the taxa (i.e., little coloured boxes, not just description in the caption text). I have great difficulties identifying the taxa based on the colour description (e.g., who would know what "heirloom corn" or "dark gold" are without looking it up). Also the x-axis label should be reformatted to a more readable text, and it also needs to be explained in the figure caption. Are the values displayed for "control" etc. averages of replicates? I am not sure how they differ from the results in Figure 2, which also shows relative abundances of in control and chlorinated samples at day 7.

- Figure 1a: In the figure caption you describe samples from "filtered" and "unfiltered" air. However, the x-axis label indicates that the "filtered air" samples were actually kept with exclusion of air/oxygen ("anox"). Can you please clarify this? Is it filtered air or is it anoxic?
- Figure 1a: The y-axis shows relative abundances in %, however, its values range from 0 to 1. This should probably be multiplied by 100, right (or the % removed), otherwise it would mean that you only show <1% of your ASVs.
- Figure 1b: I do not understand how that tree shows that the distribution is more clustered than by chance alone. This probably needs some further explanation in the figure caption.
- Figure 2: Some comments as above concerning the axis labels. Also please make sure that you do not have redundant information, could Fig. 1 and 2 maybe be combined?
- Instead, could you include a figure showing your PCoA ordination? It would be much more informative in showing how your treatments and timepoints cluster than displaying the single replicates as stacked bargraphs.
- Any statistical information (e.g., ADONIS F and R2) should be mentioned in the figure or table in which the samples are compared rather than just mentioned in the text. It is rather confusing for the reader to just have these values without seeing the actual data. E.g., in line 322 you mention F and R2 for the data shown in Fig. 3, and then you repeat the same information again in the figure caption. It's enough if it's in the figure caption.
- Figure 3: Could you please add a proper colour legend here as well? Also, you should consider using more contrasting colours for the representation of different chlorine concentrations rather than different shades of red (might not be easily distinguishable for all readers).
- Figure 3: Please clarify that the data displayed in panel A refers to 16S rRNA amplicon data and not to intl1 amplicon data.

Discussion:

When thinking about sourdough, there are 3 source where microbes might originate: Water, flour or air. In your experiment, you used sterile (?) water for the setup, thus eliminating one potential source. Did you actually analyze the flour that was used for microbial communities? Would have been interesting to see, what microbes might have originated from the flour you used. Maybe this could be discussed a little bit more?

Supplemental figures:

- Figure S2: line 649/650: The first sentence does not fit the figure. It indicates that there should be two panels (a and b), one showing general diversity and the other intl1 abundance. But you have three panels here that all show diversity estimates.
- Figure S3: Please change the colours for the different chlorination levels and provide a colour legend.

Staff Comments:

Preparing Revision Guidelines

Please return the manuscript within 60 days; if you cannot complete the modification within this time period, please contact me. If you do not wish to modify the manuscript and prefer to submit it to another journal, please notify me of your decision immediately so that the manuscript may be formally withdrawn from consideration by Microbiology Spectrum.

Dr. Gabriel G Perron
Bard College
Biology
30 Campus Road
Annandale-On-Hudson, NY 12504

Re: Spectrum01121-23 (Water chlorination increases the relative abundance of an antibiotic resistance marker in developing sourdough starters)

Dear Dr. Gabriel G Perron:

Many thanks for your patience as we found expert reviewers to assess your manuscript. You can see from the comments below that there are questions regarding the methodology and result interpretation that are important to address.

[...]

Sincerely,

Kate Howell

Journals Department
Reviewer comments:

Reviewer #2 (Comments for the Author):

The manuscript (Spectrum01121-23) investigated the effects of both surrounding air and water chlorination on bacterial communities during the development of sourdough starters. The abundance of integron 1 gene was quantified by a qPCR method in order to discuss the horizontal exchange of antibiotic-resistance genes.

Some findings were introduced in the results section. For example, the tested chlorination levels within the United States of America regulation have limited impacts on sourdough bacterial community structures. This finding seems to be informative for bakers making sourdough there.

Despite the potentially interesting findings, the conclusion is not fully supported by the data. In abstract, authors mentioned that this study indicated that water chlorination can favor the growth of key spoilage bacteria, but these key spoilage bacteria did not be explained taxonomically by 16S amplicon sequencing analysis. Thus, the authors need to show additional data to substantiate their claims. In addition, I think it is better to mention the effects of surrounding air on the bacterial community in abstract section.

We thank the Reviewer 1 for their generous comments. To answer their comments and suggestions, we change the focus of our conclusion pertaining to the presence of potential spoilage bacteria to rather focus on the presence of a genetic mechanism associated with spoiling bacteria and pathogenic bacteria. We thus updated parts of our abstract and discussion to reflect that change. We also added the mention of air quality in the abstract as well as in the introduction. Once again, we thank Reviewer 1 for raising this

issue as we believe this makes for a more accurate, and thus stronger, conclusion.

My comments and suggestion on several important points are below.

1. Materials and methods section

Line193. Does the Nano seq paired-end Illumina platform actually exist? I cannot find this platform in Illumina website. <https://jp.illumina.com/systems/sequencing-platforms.html>

Oups! Our mistake. We were working on two manuscripts at the time, one using Nano-seq (for ITS sequencing) and the current one, for which we use the more common MiSeq for 16S work. Thanks for identifying this mistake. We corrected it in the text and added a reference to the Earth Microbiome's Project protocol.

Line248. Why the int1 gene abundance was divided by the adjusted 16S rRNA copy number? I did not understand the meaning of the value. Does the value try to estimate int 1 gene abundance per one bacterial cell? Please explain it more detail. Thus, I cannot evaluate the results and discussions of the qPCR analysis.

Thanks for your question. Allow us to bring some clarification on this protocol. Because of changes in microbial count that may arise for biological reasons as well as technical reasons, it is common practice to standardize the intI1 copy numbers by dividing the latter by the 16S copy number found in the same sample. This way, we get the amount of intI1 gene relative to 16S rRNA gene, or relative abundance. We changed the sentence in the manuscript to provide clarification as follows:

“To normalize our *intI1* findings and allow comparisons between samples, we divided the *intI1* copy number by the *16S rRNA* copy number to provide us with a measured of *intI1* relative abundance for each sample.”

2. Results section

Line 282 What is the values of ADONIS: $F(1.16) = 27.7$, $R^2=0.65$, $P=0.65$, respectively? Please explain the meanings of these values in Materials and Methods section more detail.

We added the following sentence in the Materials and Methods section to provide further explanation regarding the use of ADONIS:

“The latter is a non-parametric method that estimates *F*-values from distance matrices among groups and relies on permutations to determine the statistical significance of observed differences among group means.”

Line 290. What is the meaning of "invasion"? Did the authors think lactic acid bacteria are outcompeted by the invaded bacteria? If so, please explain it more detail and discuss the mechanistic reason in discussion section.

This is a good question. To avoid using a term with a clear definition in ecology, we modified the sentence to be a more accurate description of our finding:

“ [...] we found that the starters grown in sterile conditions were significantly less likely to result in communities where lactic acid bacteria made up at least 80% of the total identified ASVs.”

Figure 1a. The relative abundance is usually shown by 100% bar chart. But, many bar charts seemed not to be 100%. Why?

As stated in the Figure Legend, this is the representation of the eight most common bacteria genera. It is common to only show relative abundance for a smaller group of taxa to facilitate visualization as it is practically very challenging to represent up to a hundred taxa.

Line 311. I cannot find figure 2A and 2B.

The order of the Figures of the Figure was modified to make sure that all figure are cited in order in the text.

Line 319. Is the unit of "µg/L" true?

Nope. The unit should be reported a “mg/L” as stated in the Materials and Methods. Thanks for identifying this typo. We made the correction in the Figure Legend and made sure we didn’t made the same mistake elsewhere in manuscript.

Line336. Please add the reference on your previous works.

We added the references as suggested. Thanks.

Line355. The results of int1 relative abundance at the day 4 are interesting. Please discuss the reason why there were no difference among different chlorine conc.

Thanks for this suggestion. We added the following clause in the Results:

“[...] day four, suggesting that selective pressures for *intI1* could be not be detected at this point.”

We also added the following in the Discussion:

“Interestingly, we did not observe this difference after four days of feeding the sourdough or between the other concentrations of free chlorine. This result suggests that the effect of free chlorine accumulates over time and is only detectable at higher concentrations found in public water systems.”

Line391. Please add the reference on the previous prediction.

Done.

3. Discussion section

Line410. I cannot understand why the whole genome sequencing is useful to evaluate the effect of water chlorination. Can this technique be applied to an environmental survey of sourdough ecosystem? I guess this technique is usually used for genome sequence of a microbial strain which is purely cultured. Please explain it more detail.

The idea here is that, as pointed out earlier, the presence of integron 1 alone cannot confirm the presence of specific spoilage or pathogenic bacteria in the sourdough starters. The use of whole-genome sequence to characterize the whole population (also know as metagenomics) in combination with metagenome-assembled-genome analyses would enable us to identify, at least partly, some of the genomic backgrounds where integron 1 are found. We modified the sentence as follows:

“Our findings also suggest that whole-genome sequencing conducted at the population level, sometimes referred to as metagenomic, might be required to fully understand the finer changes [...]”.

Reviewer #3 (Comments for the Author):

Dear authors,

thank you for this very interesting read. You have conducted a nice study on the effect of water chlorination on sourdough starters.

Whilst I liked the manuscript overall, there are multiple points that need to be addressed, most of them related to the presentation of the results. I hope that you will find these comments helpful to improve your manuscript. Please find those comments below.

We would like to thank the reviewer for their kind remark and the generous comments they made throughout the text. We hope they will be satisfied with our responses documented below. We sincerely believe our manuscript is stronger following our revisions.

One thing though: It is indicated in the text that you also have supplemental tables, which I, however, could not find anywhere. Have they been forgotten in the submission process?

Indeed, the supplemental tables have been omitted during the submission. Thanks for pointing this out. We have submitted Table S1, Table S2, and Table S3 (and made sure to mentioned Table S2 in the manuscript as well).

Material and Methods:

- Line 177: What colour chart? Was this a commercial kit?

Yes. This was part of the commercial LaMotte Chlorine Test kit. We made the mention of the kit clearer in the manuscript.

“We tested for the water chlorination level in each water preparation using the N, N-dimethylacetamide method as implemented using the LaMotte Chlorine test kit (LaMotte, Baltimore, USA).”

- Line 203pp: Please state what parameters were used for the dada2 and following analyses. E.g., were standard parameters used or did you modify them in any way?

Good point! We modified our Materials and Methods as follow:

“More specifically, we processed the *16S rRNA* reads using the *DADA2* pipeline version 1.20 (Callahan et al., 2016) available at (<https://github.com/benjjneb/dada2>) using standard parameters unless specified and implemented in *R* version [...]”.

We also modified this section by adding some elements of the Results section as suggested below:

“In total, we obtained a total of 1,160,000 pairs of forward and reverse reads, excluding eight samples that failed to sequence) with an average read length of 250 base pairs, totaling ~583 G bases and an average sequencing depth per sample of 41,642.9 paired-reads. Each sequence read were then quality-checked, trimmed (i.e., forward reads at 240 bp and reverse reads at 225 bp), assessed for chimeric contaminants, and de-noised for possible sequencing error. After processing the reads for quality-check, we were conserved 980,261 (84.1% of the initial) paired-reads.”

- Line 218pp: Did you use rarefaction or any other means of normalization to counteract the potential effect of uneven read-counts on diversity estimates?

Yes. We stated that we subsampled each sample (i.e., rarefied) in the Material and Methods section, but it wasn't very clear. We modified the following sentence as follows:

“To estimate diversity indices, we rarefied all samples to the lowest sampling depth, and estimated the total number of [...]”.

Results

- Line 264-266: This information about trimming and removal of chimeric reads should go to the material and methods section (see previous comments about analysis parameters).

See above.

- Line 279: What was the idea behind limiting air exposure? Did you want to avoid bacteria from the air getting into your starter? Or was your main target the exclusion of oxygen from the incubation? This should be stated somewhere. In line 289, you refer to "sterile conditions" so was your target the exclusion of microbes from the air?

This is a good question, which was not properly addressed in our manuscript. While the focus of our study was the effect of chlorination on microbial communities, the results regarding air quality ended up

being fairly interesting. The idea here was to test the effect of chlorination in a clean environment vs a “normal” environment, with the hypothesis that the clean environment would not be suitable for sourdough culture. As suggested by another reviewer (see above), we made our “air” results more obvious and clearly started throughout the text (e.g., abstract, intro, and discussion). We also modified the result as follows:

“We first wanted to identify the ideal condition to establish sourdough starters in our facilities. To do so, we established two sourdough starters trials: one under sterile laboratory conditions with filtered air and the second in a kitchen environment with exposure to unfiltered air.”

- Figure 1a: Please include a figure legend which maps the bar colours to the taxa (i.e., little coloured boxes, not just description in the caption text). I have great difficulties identifying the taxa based on the colour description (e.g., who would know what "heirloom corn" or "dark gold" are without looking it up). Also the x-axis label should be reformatted to a more readable text, and it also needs to be explained in the figure caption. Are the values displayed for "control" etc. averages of replicates? I am not sure how they differ from the results in Figure 2, which also shows relative abundances of in control and chlorinated samples at day 7.

We added a color legend in the figure. It is absolutely true that it is hard to identify the different taxa with the description in the legend only (even though many journals used to request not having color legend as part of a panel; time to move on!).

- Figure 1a: In the figure caption you describe samples from "filtered" and "unfiltered" air. However, the x-axis label indicates that the "filtered air" samples were actually kept with exclusion of air/oxygen ("anox"). Can you please clarify this? Is it filtered air or is it anoxic?

Thanks for point this out. We had used anoxic as a shortcut for our samples as in one early trial we did test anoxic conditions, but we had decided against it for our main set of experiments. We corrected this mistake in the new legend.

- Figure 1a: The y-axis shows relative abundances in %, however, its values range from 0 to 1. This should probably be multiplied by 100, right (or the % removed), otherwise it would mean that you only show <1% of your ASVs.

Corrected. Thanks!

- Figure 1b: I do not understand how that tree shows that the distribution is more clustered than by chance alone. This probably needs some further explanation in the figure caption.

After carefully reviewing the purpose of Figure 1 and 2 as suggested by Reviewer #3, we decided to modify the figures quite substantially. Figure 1A remains more or less the same with small edits to improve clarity. Figure 1B is presenting graphically a NMDS analysis of the populations structure. The differences between the two groups is much more obvious now.

Then, the original Figure 2 was changed to Figure S3 with a few edits to improve clarity. As the reviewer pointed out, there was a non negligible overlap between Figure 1 and 2 previously in terms of novel information. Now, Figure 2 show the changes in predicted ASVs and diversity in the experiment, which present novel information altogether.

Thanks once again for making us think over the figure presentation!

- Figure 2: Some comments as above concerning the axis labels. Also please make sure that you do not have redundant information, could Fig. 1 and 2 maybe be combined?

See above.

- Instead, could you include a figure showing your PCoA ordination? It would be much more informative in showing how your treatments and timepoints cluster than displaying the single replicates as stacked bargraphs.

See above.

- Any statistical information (e.g., ADONIS F and R2) should be mentioned in the figure or table in which the samples are compared rather than just mentioned in the text. It is rather confusing for the reader to just have these values without seeing the actual data. E.g., in line 322 you mention F and R2 for the data shown in Fig. 3, and then you repeat the same information again in the figure caption. It's enough if it's in the figure caption.

We added information pertaining to the statistical analysis in the Figure Legend as suggested. We also kept the statistical information in the manuscript as we believe it is a key information that is used to support/construct our results section. It would be similarly confusing to the authors to have to go back and forth to the Legend to understand whether a claim is statistically supported.

- Figure 3: Could you please add a proper colour legend here as well? Also, you should consider using more contrasting colours for the representation of different chlorine concentrations rather than different shades of red (might not be easily distinguishable for all readers).

We updated the colors.

- Figure 3: Please clarify that the data displayed in panel A refers to 16S rRNA amplicon data and not to int11 amplicon data.

Actually, we remove Panel A) and focus entirely on the results regarding the int11 relative abundance. This avoid redundancy between Figure 2 and Figure 3.

Discussion:

When thinking about sourdough, there are 3 source where microbes might originate: Water, flour or air. In your experiment, you used sterile (?) water for the setup, thus eliminating one potential source. Did you actually analyze the flour that was used for microbial communities? Would have been interesting to see, what microbes might have originated from the flour you used. Maybe this could be discussed a little bit more?

This is a very good point. It turns out that we tried to extract DNA and sequence the microbial communities present in the flour twice, but both samples failed to sequence properly, giving very low counts and poor quality. The samples were thus taken out from our analysis. That being said, the fact that we used sterile water and sterile air suggests to use that the microbial communities developing in the latter starters most likely originated from the flour as the sole source of microbial contaminants. To this

end, we found *Pantaea* spp. as a major taxon in the sterile air starters, which was previously found as a common contaminant of raw flour. We discussed this in our discussion (see lines 404-411).

Supplemental figures:

- Figure S2: line 649/650: The first sentence does not fit the figure. It indicates that there should be two panels (a and b), one showing general diversity and the other intl1 abundance. But you have three panels here that all show diversity estimates.

Wow! How did this happen? We updated the figure to reflect what we intended to show. Thanks!

- Figure S3: Please change the colours for the different chlorination levels and provide a colour legend.

We actually modified Figure S3 altogether.

Re: Spectrum01121-23R1 (Water chlorination increases the relative abundance of an antibiotic resistance marker in developing sourdough starters)

Dear Dr. Gabriel G Perron:

Your manuscript has been accepted, and I am forwarding it to the ASM production staff for publication. Your paper will first be checked to make sure all elements meet the technical requirements. ASM staff will contact you if anything needs to be revised before copyediting and production can begin. Otherwise, you will be notified when your proofs are ready to be viewed.

Sincerely,
Kate Howell
Editor
Microbiology Spectrum

Reviewer #2 (Comments for the Author):

The manuscript has been revised well according to reviewer comments.